# Aspartame Consumption, Mitochondrial Disorder-Induced Impaired Ovarian Function, and Infertility Risk

**DOI:** 10.3390/ijms232112740

**Published:** 2022-10-22

**Authors:** Yang-Ching Chen, Yen-Chia Yeh, Yu-Fang Lin, Heng-Kien Au, Shih-Min Hsia, Yue-Hwa Chen, Rong-Hong Hsieh

**Affiliations:** 1School of Nutrition and Health Sciences, College of Nutrition, Taipei Medical University, Taipei 110, Taiwan; 2Graduate Institute of Metabolism and Obesity Sciences, Taipei Medical University, Taipei 110, Taiwan; 3Department of Family Medicine, Taipei Medical University Hospital, Taipei Medical University, Taipei 110, Taiwan; 4Department of Family Medicine, School of Medicine, College of Medicine, Taipei Medical University, Taipei 110, Taiwan; 5Department of Obstetrics and Gynecology, School of Medicine, College of Medicine, Taipei Medical University, Taipei 110, Taiwan or; 6Department of Obstetrics and Gynecology, Taipei Medical University Hospital, Taipei Medical University, Taipei 110, Taiwan; 7Research Center of Nutritional Medicine, College of Nutrition, Taipei Medical University, Taipei 110, Taiwan

**Keywords:** aspartame, infertility, mitochondria, oocyte maturation, steroidogenesis

## Abstract

Frequent consumption of diet drinks was associated with oocyte dysmorphism, decreased embryo quality, and an adverse effect on pregnancy rate. We investigated the harmful effects of aspartame and potential mechanisms through which it increases infertility risk through clinical observations and in vivo and in vitro studies. Methods: We established a cohort of 840 pregnant women and retrospectively determined their time to conceive. We assessed the estrus cycle, the anti-Mullerian hormone level, ovarian oxidative stress, and ovarian mitochondrial function in an animal study. We also evaluated mitochondria function, mitochondrial biogenesis, and progesterone release with in vitro studies. Aspartame consumption was associated with increased infertility risk in the younger women (Odds ratio: 1.79, 95% confidence interval: 1.00, 3.22). The results of the in vivo study revealed that aspartame disrupted the estrus cycle and reduced the anti-Mullerian hormone level. Aspartame treatment also suppressed antioxidative activities and resulted in higher oxidative stress in the ovaries and granulosa cells. This phenomenon is caused by an aspartame-induced decline in mitochondrial function (maximal respiration, spare respiratory capacity, and ATP production capacity) and triggered mitochondrial biogenesis (assessed by examining the energy depletion signaling-related factors sirtuin-1, phosphorylated adenosine monophosphate-activated protein kinase, peroxisome proliferator-activated receptor-gamma coactivator-1α, and nuclear respiratory factor 1 expression levels). Aspartame may alter fertility by reserving fewer follicles in the ovary and disrupting steroidogenesis in granulosa cells. Hence, women preparing for pregnancy are suggested to reduce aspartame consumption and avoid oxidative stressors of the ovaries.

## 1. Introduction

The consumption of non-nutritive sweeteners (NNSs) has markedly increased during the past two decades because of recommendations to reduce sugar intake [1]. According to a large-scale nutrition survey conducted in the United States, 25.1% of children and 41.1% of adults consume NNSs every day [1], and the consumption rate of NNSs in Asian countries is increasing recently, both in young adults and children [2]. A study reported that NNS intake reached 78% and 246% of the acceptable daily intake (ADI) for adults and children in China, respectively [2]. In addition, women consume a higher level of NNSs than do men [1], possibly due to a higher preference for sweet food among women [3]. A survey conducted in Asian countries reported that women of reproductive age consumed a relatively high level of NNSs, and most sources were NNS beverages and preserved fruits [3]. Studies have reported the adverse effects of long-term consumption of NNSs on women of reproductive age, including the increased incidence of preterm delivery [4], type 2 diabetes mellitus [5], and hypertension [6].

Aspartame is the most widely used NNS worldwide [7] and is frequently used in foods, medications, and carbonated soft drinks. Despite its widespread use, the safety of aspartame remains controversial [8]. Some recent studies have implicated the consumption of aspartame as being associated with impacts on human or rat physiological responses such as glucose intolerance [9], body fat distribution [10], and liver damage [11]. Although studies have suggested the adverse effects of aspartame consumption, they were primarily based on animal experiments, and data from human studies are scant [12]. Aspartame and its metabolites, including 10% methanol, 40% aspartic acid, and 50% phenylalanine [8], may disrupt the oxidant–antioxidant balance, induce oxidative stress, and damage cell membrane integrity. Although the safety of aspartame consumption has been evaluated and its ADI established before being marketed, few studies have investigated its effect on fertility outcomes. A study reported that the frequent consumption of diet drinks was associated with oocyte dysmorphism, decreased embryo quality, and a mild adverse effect on the pregnancy rate [13]. However, this study did not quantify the amount and type of sweeteners consumed. A study examining the effects of artificial sweeteners indicated no association between aspartame consumption and sperm quality [14]. No study has evaluated the safety of aspartame consumption in women of reproductive age who are preparing for pregnancy. Given the high prevalence of infertility in the recent two decades [15], possible toxic exposures from dietary intake that can lead to infertility risk should be urgently investigated.

Ovarian function determines a large part of fertility. Granulosa cells, which surround ovarian follicles, proliferate and secrete several steroid and peptide hormones to regulate follicular growth, development, and maturation [16]. Mitochondria present within granulosa cells and oocytes are the leading site for ovarian follicle steroidogenesis, which is crucial for the formation of follicular cells or the transformation of fertilized eggs to blastocysts [17]. Hence, the mitochondrial dysfunction of human granulosa cells may contribute to decreases in steroidogenesis, fertilization rate, oocyte maturation rate, and oocyte quality, ultimately causing infertility [18]. The mitochondria balance oxidative stress within the cytoplasm [19]. A Wistar rat experiment reported that aspartame consumption increases oxidative stress in the heart [20], liver, and kidneys. However, no experimental study has investigated oxidative stress induced by aspartame in the female reproductive system, in either human or animal studies [21].

Following the scientific gaps addressed in previous paragraphs, this study investigated whether aspartame consumption increases the risk of infertility in a longitudinal cohort of pregnant women. The effect of aspartame consumption on time to conceive and infertility risk was further stratified by age. In addition, to confirm the phenomenon observed in the clinical setting, we examined the effect of different aspartame dosages on the function of the mitochondria and the biogenesis of the ovaries by using an animal model. Further, the animal model was also used to determine whether aspartame consumption modifies fasting glucose, insulin, triglycerides, and transaminases. In addition, human granulosa-like tumor cells (KGN cells) were used to explore pathological conditions, namely mitochondrial oxidative stress, mitochondrial function, and progesterone secretion, under different concentrations of aspartame exposure.

## 2. Results

### 2.1. Clinical Characteristics of Study Participants

A total of 840 pregnant women were enrolled, including 164 women (19.52%) with infertility (Table 1). The mean age of the participants was 33.6 years (age ranged from 19 to 49) and half of them were aged <35 years. Furthermore, 10.25% and 6.08% of the women were overweight and obese, respectively, according to their pre-pregnancy BMI. Nearly one-fourth of the participating women habitually consumed aspartame.

### 2.2. Relationship between Aspartame Consumption and Infertility Risk

Among the 840 participants, aspartame consumption was associated with an increased risk of infertility with an odds ratio (OR) of 1.30 (95% confidence interval [CI]: 0.87, 1.93; Table 2). This association was near significant in the women aged <35 years (OR: 1.79, 95% CI: 1.00, 3.22). After further categorizing the women aged <35 years into low and high consumption groups, the OR of infertility increased from OR: 1.58 (95% CI: 0.73, 3.44) among low aspartame consumption to OR: 2.01 (95% CI: 0.97, 4.17) among high aspartame consumption; a significant dose–response association was observed (*p* for trend = 0.04, Figure 1). Time to conceive was longer in the women who consumed aspartame than in those who did not (β = 1.28 month, 95% CI: −1.34, 3.91).

### 2.3. Aspartame Caused Higher Insulin Resistance and Lipid Abnormal Accumulation

To examine the mechanism underlying the effect of aspartame on female fertilization, we use rat animal models were given aspartame for 12 weeks and KGN cells were treated with 5–20 Mm aspartame for 48 h, respectively. No significant change in body weight and food intake was noted among the experimental groups. The rats that were given aspartame for 12 weeks had significantly increased serum fasting glucose, insulin, HOMA-IR, AST, and ALT levels (Table 3).

Aspartame intervention affected the lipid profile and caused kidney fat accumulation. Higher levels of serum triglycerides and kidney fat were found in the aspartame group (Table 3).

### 2.4. Aspartame-Induced Oxidative Stress in the Ovary and Granulosa Cells

Decreased expression levels of CAT (Figure 2A) and SOD-2 (Figure 2B) were observed in the ovaries of the aspartame-treated rats. The levels of oxidative stress markers, namely 8-OHdG and MDA, were increased in the ovaries of the aspartame-treated rats (Figure 2C,D). The cellular ROS levels in the groups treated with 10 and 20 mM aspartame were significantly increased (Figure 2E). These findings indicated that aspartame treatment may suppress antioxidative activities and result in higher oxidative stress in many organs including the ovaries.

### 2.5. Aspartame-Compromised Mitochondrial Function and Triggered Mitochondrial Biogenesis

Mitochondrial respiratory function including NCCR (mitochondrial complex I and III) and SCCR (mitochondrial complex II and III) activities was decreased in both aspartame treatment groups (Figure 3A). The dysfunction of mitochondrial respiratory function may result from higher oxidative stress caused by aspartame.

The expression levels of the energy depletion signaling-related factors SIRT1, pAMPK, PGC-1α, and NRF1 were increased in the ovaries of the aspartame-treated rats (Figure 3B). In addition, the mitochondrial DNA copy number was higher in the ovaries of the aspartame-treated rats (Figure 3C). These data indicate that mitochondrial biogenesis was induced to compensate for mitochondrial dysfunction.

Similarly, significantly decreased OCRs for maximal respiration, spare respiratory capacity, and adenosine triphosphate (ATP) production capacity were observed in the aspartame-treated granulosa cells (Figure 3D). Increased protein levels of ATP5a, PGC-1α, nuclear respiratory factor 1, and TFAM were noted after aspartame treatment for 48 h (Figure 3E).

### 2.6. Aspartame Altered Fertility by Reserving Fewer Follicles in the Ovary and Affecting Steroidogenesis in Granulosa Cells

The results of the vaginal smear examination indicated that the diestrus period was prolonged after aspartame treatment (Figure 4A). In addition, decreased levels of AMH were determined in the rats after 10 weeks of aspartame treatment (Figure 4B,C).

Steroidogenesis function requires ATP hydrolysis and an electrochemical gradient for maximal steroidogenic activity. The granulosa cells treated with 20 mM aspartame exhibited significantly decreased progesterone secretion, which is highly correlated with ATP production (Figure 4D).

## 3. Discussion

Our study results revealed that aspartame increased oxidative stress in the reproductive system, including the granulosa cells and ovaries, reduced mitochondrial function, and boosted the compensatory mechanism to increase mitochondrial biogenesis. However, the compensatory mitochondrial biogenesis process could not recover from the mitochondrial dysfunction caused by aspartame, leading to an increased risk of infertility.

No significant change was noted in the body weight of the aspartame treatment group in this study. However, the inguinal fat weight/body weight ratio of the aspartame group was significantly higher than that of the control group. Several animal studies have reported that aspartame did not affect body weight; however, it affected body composition and increased body fat in rats and mice experiments [10,22,23]. Significantly increased fasting glucose, insulin, and HOMA-IR levels were observed with no difference in food intake, indicating that aspartame may upregulate endogenous glucose production (EGP) independently of its low energy. Gul et al. [24] reported that aspartame may inhibit alkaline phosphatase activity in the intestine to promote glucose intolerance. Palmnäs et al. indicated that aspartame significantly increased fasting blood glucose and propionate production in the colon in male SD mice [22], and propionate is a gluconeogenesis substance in the liver [25]. In the postabsorptive state, EGP metabolized levels were identical in the liver and kidney [26]. With an increase in the fasting condition, EGP through renal gluconeogenesis increased [27]. Long-term EGP effects may promote triglyceride synthesis and kidney fat accumulation, as observed in this study.

Aspartame may induce oxidative stress through both decreased antioxidative enzyme expression and increased reactive oxygen species production, which were determined in the ovary and granulosa cells. Fimanor et al. presented that malondialdehyde lipid peroxide in the liver was significantly increased and the activities of SOD-2, CAT, and glutathione peroxidase were significantly decreased, leading to an imbalance of oxidative stress after aspartame intervention for 12 weeks [11]. Similar effects were observed in the heart [20]. Our results are consistent with those of previous studies. The oxidative stress index results indicated that the levels of both 8-hydroxydeoxyguanosine and malondialdehyde lipid peroxide in the ovaries were increased.

Although mitochondria are the center of cellular energy production, they are the main source of ROS. ROS is a byproduct of the electron leakage of mitochondrial electron transport chain enzyme complexes I and III. Under normal circumstances, ROS is removed by the antioxidative enzymes superoxide dismutase and glutathione in the mitochondria. However, excessive ROS production damages mitochondria, resulting in increased ROS production. Zorov et al. proposed that ROS induces the release of ROS [28], increasing oxidative pressure. Increased oxidative stress causes mitochondrial dysfunction [29]. This study demonstrated mitochondrial enzyme dysfunction in both NCCR and SCCR, and compromised ATP production ability in the aspartame-treated granulosa cells and ovaries. Mitochondrial dysfunction includes the decreased activity of mitochondrial electron transport chain enzyme complexes and the loss of MMP, resulting in the inhibition of the main function of mitochondria-energy production ATP [30]. 

Mitochondrial biogenesis was induced to compensate for aspartame-induced oxidative damage resulting in dysfunctional mitochondria. Bouchez et al. reported that different species perform different mitochondrial biosynthesis regulatory mechanisms for ROS produced by mitochondria. In yeast, ROS can downregulate the pathway of mitochondrial biosynthesis through the action of Hap4p; in mammals, ROS can trigger PGC-1α and induce enhanced mitochondrial biosynthesis [31]. Previous studies have reported that in mammalian cells, mitochondrial biosynthesis is mainly performed through the NRF1 pathway [32]. NRF1 is a transcription factor that regulates the transcription of the mitochondrial gene body TFAM. Mitochondrial dysfunction results in the loss of respiratory enzyme complex activities, leading to both elevated oxidative stress and decreased ATP production. Energy deprivation produces stress factors that enhance the AMPK-mediated phosphorylation of PGC-1α, as well as increase the ratio of NAD to NADH, resulting in SIRT1 activation and PGC-1α deacetylation. Activated PGC-1α triggers Nrf and Tfam and subsequently promote mitochondrial biogenesis. In this study, we observed increased levels of SIRT1, pAMPK, PGC-1α, and NRF1, and elevated mtDNA copy numbers in the ovaries of the aspartame-treated rats. These data indicate that mitochondrial biogenesis was induced to compensate for aspartame-induced mitochondrial dysfunction. However, mitochondrial biogenesis could not rescue mitochondrial dysfunction in this study.

The estrous cycle refers to the reproductive cycle in rodents, which corresponds to the human menstrual cycle. Metestrus and diestrus are similar to human early and late secretory stages, respectively. High progesterone levels are associated with both metestrus and diestrus phases. In this study, a longer diestrus phase and reproductive dysfunction were observed in the aspartame-treated adult female rats. Reproductive dysfunction was indicated by the lower and higher occurrences of estrus and diestrus phases in neonatal overfeeding female rats [33]. AMH produced by the granulosa cells of primary and early-stage antral follicles is used to diagnose reproductive dysfunction, such as ovulatory disorders [34]. Higher insulin levels compromise granulosa cell receptivity and AMH production ability [35]. Granulosa cells are steroid-producing cells that play crucial roles in regulating oocyte maturation and early development. Mitochondria are critical organelles for steroidogenesis. Studies have reported that MMP disruption, decreased ATP synthesis, or compromised mitochondrial function may attenuate granulosa cell steroidogenesis [17,36,37].

Our human cohort study was limited by its observational nature, which could be biased by many unmeasured confounding factors. However, clinical trials are not feasible and unethical to examine our hypothesis, given the suspicious food safety issue addressed in this study. Another limitation is that we used the retrospective recall of the pregnant women regarding time to conceive, which is less accurate compared to medical records. However, for routine medical records, time to conceive is less frequently recorded and is also obtained from the womens’ self-recall.

## 4. Materials and Methods

### 4.1. Study Participants

In July 2018, we established the Taipei Mother–Infant Nutrition Cohort [38] and longitudinally followed up the birth cohort in Taiwan. We enrolled pregnant women who visited the obstetric clinics of four Taipei hospitals for prenatal care after obtaining their written informed consent. Pregnant women with severe diseases (e.g., heart diseases and cancer) and organic infertility causes (e.g., tubal factors and endometriosis) were excluded. Until May 2021, we included 1170 pregnant women who completed the baseline survey, provided dietary intake data, and completed a validated food frequency questionnaire (FFQ) to evaluate the intake of NNSs. We excluded 330 pregnant women who were unable to recall their time to conceive. Finally, we included a cohort of 840 naturally conceived women in the analysis. Gestational weight gain was determined by subtracting body weight (kg) at last prenatal visit just prior to delivery by pre-pregnancy body weight (kg) according to the medical record. This study was approved by the Joint Institutional Review Board of Taipei Medical University (N201806009 and N202001013).

### 4.2. Assessment of Aspartame Consumption

Aspartame is hydrolyzed into aspartic acid, phenylalanine, and methanol after consumption; hence, no valid biomarkers from urine or stool can adequately reflect its intake. Therefore, aspartame consumption was examined using our previously developed non-nutritive sweeteners–food frequency questionnaire (NNS–FFQ) [39]. The NNS–FFQ was designed to capture the habitual dietary intake of NNSs in the past 3 months, mimicking the status of long-term NNS consumption. Intensive and overall market research was performed to develop the applicable NNS–FFQ with 13 food categories and 305 items. Aspartame consumption was calculated by summarizing the intake of 32 food items, considering the portion size and frequency. Total calorie intake was estimated from an image-based 3-day dietary recall [40]. 

### 4.3. Assessment of Fertility Outcomes and Covariates

We used a participant-reported questionnaire at the baseline survey to define the time to conceive. The question was as follows: “How long have you been preparing for this pregnancy?” Infertility was defined as time to conceive longer than 12 months even under normal sexual life and preparation for pregnancy period. Covariates that were reported to influence sweeteners intake and reproductive outcomes, such as maternal age, educational level, family income, pre-pregnancy obesity status, and total calorie intake, were also assessed.

### 4.4. Animal Model

Sprague Dawley rats (weighing 200–220 g), obtained from BiOlasco (Taipei, Taiwan), were kept two to three per cage at a constant humidity of 50–60% and a temperature of 23 °C ± 2 °C under a 12 h dark/light cycle. They were fed Laboratory Rodent Diet 5001 and water ad libitum. The experimental protocol was approved by the Animal Research Committee (Institutional Animal Care and Use Program, IACUC) of Taipei Medical University (IACUC Approval No: LAC-2020-0176, LAC-2020-0278).

Following 1-week acclimatization, rats were divided into three groups (n = 10): (i) control, receiving 1% carboxymethyl cellulose; (ii) those receiving a low dose of aspartame, at a dose of 30 mg/kg; and (iii) those receiving a high dose of aspartame, at a dose of 60 mg/kg. Aspartame was purchased from Alfa Aesar. The dose of aspartame (30 or 60 mg/kg; 1 mL per rat, prepared in 1% carboxymethyl cellulose) or vehicle was administered using a gavage for 12 weeks. The dose of aspartame for rats were determined according to average consumption dose from Asian population [41] for the low dose group and referencing the European population consumption for the high dose group [42]. The amount of food consumed did not significantly differ among the groups.

After 12 weeks, the animals from each group were further equally divided into two groups (n = 5): (i) nonpregnant: control, low dose of aspartame, and high dose of aspartame; and (ii) pregnant: control, low dose of aspartame, and high dose of aspartame. In week 13, the nonpregnant group was fasted overnight and anesthetized with 5% isoflurane. Their blood was collected and ovary tissues were removed and weighed. By contrast, the pregnant group was mated with male rats at a ratio of 1:1. Presence of sperm in the vaginal smear was defined as day 1 of pregnancy, and each rat was kept in a separate cage. On the 18th day of gestation, the pregnant rats were fasted overnight and anesthetized with 5% isoflurane. Their blood was collected, and ovarian tissues were removed, weighted, and subjected to histological examination. All the samples were frozen at −80 °C until further analysis.

### 4.5. In Vitro Cell Culture Experiment

KGN (RIKEN BRC Cat# RCB1154, RRID:CVCL_0375) granulosa cells were maintained in Dulbecco’s modified Eagle medium/nutrient mixture F-12 (Sigma-Aldrich, St. Louis, MO, USA) and supplemented with 10% fetal bovine serum (GeneTeks Bioscience, Taipei, Taiwan) and 1% antibiotics (100 units/mL penicillin, 0.1 μg/mL streptomycin, and 0.25 μg/mL amphotericin) in a humidified atmosphere of 5% CO_2_ at 37 °C.

Aspartame was purchased from Alfa Aesar GmbH & Co. KG (Ward Hill, MA, USA), and stock solutions of aspartame (100 mM) were prepared in phosphate-buffered saline (PBS; 137 mM NaCl, 2.7 mM KCl, 12 mM NaHPO_4_, and 2 mM KH_2_PO_4_). The stock solutions were filtered using a 0.22-μm filter before use in the experiment. The stock solution was diluted with a serum-free complete medium to prepare working concentrations. In the pilot KGN cell experiment, 10 mM aspartame decreased cell viability, while 20 mM increased cell viability. From the preliminary results, we speculated that 10 and 20 mM aspartame may affect mitochondrial activity in cells without causing significant cytotoxicity (Figure A1). Hence, we determined to treat the KGN cell with 10 and 20 mM aspartame.

### 4.6. Glucose, Insulin, Triglycerides, and Transaminases

All the blood samples were collected in estrus after fasting overnight. Blood was centrifuged at 3000 rpm for 10 min at 4 °C. Serum glucose and triglyceride levels were examined using a commercial kit (Randox, Crumlin, UK) based on the enzymatic colorimetric method. Insulin levels were examined using the enzyme-linked immunosorbent assay (ELISA) kit (Mercodia, Uppsala, Sweden). Aspartate aminotransferase (AST) and alanine aminotransferase (ALT) were detected using a kit based on the kinetic method (Teco Diagnostics, CA, USA). Insulin resistance was diagnosed using the homeostatic model assessment for insulin resistance (HOMA-IR) formula. The HOMA-IR index was calculated as fasting insulin (mIU/L) × fasting glucose (mmol/L)/22.5.

### 4.7. Assessment of Changes in Anti-Mullerian Hormone Levels and Estrus Cycle

The estrus cycle was examined daily between 10:00 and 12:00 through the microscopic examination of a vaginal smear before blood sample collection, mating, and sacrificing to determine the proestrus cycle. Vaginal cytology was classified into four stages (proestrus, estrus, metestrus, and diestrus) in accordance with the findings of previous studies [43,44]. Cycle data were analyzed in weeks 8 to 12.

All blood samples for examining the anti-Mullerian hormone (AMH) level were collected in estrus after fasting overnight. The blood samples were centrifuged at 3000 rpm for 10 min at 4 °C. AMH levels were measured using an enzymatically amplified two-sided immunoassay (Cusabio, TX, USA). Results are expressed as milligrams per deciliter.

### 4.8. Ovarian Oxidative Stress

We examined the level of 8-hydroxy-2′-deoxyguanosine (8-OHdG), a major product of DNA base oxidation, by using the sandwich enzyme-linked immunosorbent assay (Cusabio, Fannin). Results are expressed as nanograms per milliliter.

We examined the levels of thiobarbituric acid reactive substances (TBARSs), a marker of oxidative damage derived from lipid peroxidation, in the ovary tissues by using a commercial kit (Cayman, MI, USA). First, the supernatant (10 μL) of the tissue homogenate was mixed with 10 μL sodium dodecyl sulfate (SDS) and 400 μL color reagent. Subsequently, the samples were incubated for 60 min at 95 °C. After cooling, the precipitate was removed through centrifugation at 1600× *g* for 10 min at 4 °C. The reaction product was measured through spectrophotometry at 532 nm. Results are expressed as nanomoles per milligram of protein.

### 4.9. Ovary Mitochondrial Function Assessment

To determine nicotinamide adenine dinucleotide (NAD) phosphate-cytochrome c reductase (NCCR) and succinate cytochrome c reductase (SCCR) activity, 180 μL of NCCR test solution (1 mM NADH, 1.5 mM potassium cyanide, 50 mM potassium phosphate buffer, pH 7.4), and SCCR test solution (25 mM succinate, 1.5 mM potassium cyanide, 50 mM potassium phosphate buffer, pH 7.4) were added to 20 μg of mitochondrial extraction and incubated for 2 min at 37 °C. Subsequently, 20 μL of 1 mM oxidized cytochrome c was added. The activity was measured through spectrophotometry for 10 min at 550 nm. Results are expressed as nanomole per cytochrome c reduced per minute per milligram of protein.

### 4.10. Quantification of mtDNA

mtDNA was extracted from the ovaries by using the DNeasy blood and tissue kit (Qiagen, USA). To determine the relative mtDNA copy number, the levels of mitochondrial NADH-CoQ oxidoreductase 1 (ND-1) relative to standard β-actin were evaluated. The following primers were used: ND-1 (forward: 5′-TTA ATT GCC ATG GCC TTC CTC ACC-3′ and reverse: 5′-TGG TTA GAG GGC GTA TGG GTT CTT-3′ [21]) and β-actin (forward: 5′-ACA GGA TGC AGA AGG AGA TTA C-3′ and reverse: 5′-ACA GTG AGG CCA GGA TAG A-3′). Relative transcript abundance was calculated using the delta–delta cycle threshold (∆∆Ct) method.

### 4.11. Western Blotting

The frozen ovary samples were homogenized in extraction buffer (0.2% SDS, 0.2% DOS, 1% Triton X-100, 50 mM Tris-HCl, 1 mM ethylenediaminetetraacetic acid, and 5% protease inhibitor cocktail) on ice. Homogenates were centrifuged at 10,000 rpm for 15 min at 4 °C. The supernatants were used in the assay. Protein extracts (30 μg) were separated through 10% SDS-polyacrylamide gel electrophoresis, transferred onto polyvinylidene fluoride membranes, and probed with antibodies against NAD-dependent deacetylase sirtuin-1 (SIRT1; CST, MS, USA), adenosine monophosphate (AMP)-activated protein kinase (AMPK; Abcam, Cambridge, UK), peroxisome proliferator-activated receptor-gamma coactivator (PGC-1α; Novus, CO, USA), nuclear respiratory factor 1 (NRF1; CST, MS, USA), mitochondrial transcription factor A (TFAM; Abcam, Cambridge, UK), catalase (CAT; Biolegend, CA, USA), superoxide dismutase (SOD; CST, MS, USA), and glyceraldehyde 3-phosphate dehydrogenase (GAPDH; Sigma-Aldrich, MA, USA)

### 4.12. Determination of Intracellular and Mitochondrial Reactive Oxygen Species Levels

Cellular and mitochondrial reactive oxygen species (ROS) were measured using 2′,7′-dichlorofluorescein diacetate (DCFDA; Cayman Chemical, MI, USA, Ex/Em = 488 nm/530 nm) and mitochondrial ROS detection reagent (Cayman Chemical, MI, USA, Ex/Em = 488 nm/574 nm), respectively. Antimycin A (Sigma-Aldrich, MA, USA), an inhibitor of respiration complex III, was used as a positive control to induce ROS production. N-acetylcysteine (NAC), a precursor of the antioxidant glutathione, was used as a negative control to reduce ROS production. KGN cells were treated with different concentrations of aspartame (5, 10, and 20 mM) for 24 and 48 h. At the end of treatment, the cells were removed from the dish and centrifuged at 400× *g* for 2 min. After the addition of 5 μM DCFDA to each tube, 21 mM NAC was added to the negative control at 37 °C in the dark for 30 min, followed by 30 min incubation. Subsequently, 10 μM antimycin-A was added as the positive control, and the mixture was incubated for an additional hour at 37 °C in the dark. To determine the mitochondrial ROS level, mitochondrial ROS detection reagent was added to each tube, followed by incubation for 20 min at 37 °C in the dark, and the cells were washed twice with Hank’s balanced salt solution (HBSS; ScienaCell, CA, USA). Subsequently, 10 μM Antimycin-A was added as the positive control, and the mixture was incubated for an hour at 37 °C in the dark. Finally, both DCFDA and mitochondrial ROS detection reagents were removed and resuspended in HBSS. All the tubes were kept in the dark until flow cytometry analysis (Attune NxT flow cytometer, Thermo Fisher).

### 4.13. Determination of Mitochondrial Membrane Potential and Mitochondrial Mass

Mitochondrial membrane potential (MMP) and mitochondrial mass were examined using tetramethylrhodamine ethyl ester (TMRE; Cayman Chemical, MI, USA, Ex/Em = 488 nm/574 nm) and MitoView Green (Biotium, CA, USA, Ex/Em = 488 nm/530 nm). Carbonyl cyanide 4-(trifluoromethoxy) phenylhydrazone (FCCP; Sigma-Aldrich, MA, USA), a mitochondrial oxidative phosphorylation uncoupler, was used as a negative control to reduce MMP. The KGN cells were treated with different concentrations of aspartame (5, 10, and 20 mM) for 24 and 48 h. At the end of treatment, the cells were removed from the dish and centrifuged at 400× *g* for 2 min. TMRE and MitoView Green were added to each tube to achieve final concentrations of 100 nM TMRE and 100 nM MitoView Green, respectively, and the tubes were incubated in the dark for 30 min. The fluorescent dye was removed and resuspended in PBS. All the tubes were kept in the dark until flow cytometry analysis (Attune NxT flow cytometer, Thermo Fisher).

### 4.14. Oxidative Phosphorylation Assay for Mitochondrial Function

Cellular oxidative phosphorylation (OXPHOS) was monitored using a Seahorse XFe24 extracellular flux analyzer (Seahorse Bioscience, MA, USA) in real time by measuring the oxygen consumption rate (OCR). In brief, 6 × 104 cells were seeded onto 24-well plates for XFe24 in 100 μL of serum-free medium and incubated with 5% CO_2_ for 6–7 h when the cells were attached to the wells. Before obtaining the measurements, we immersed the cells in 625 μL of the assay medium, incubated them at 37 °C for 1 h in the absence of CO_2_, and serially stimulated them in the following sequence: (A) 40 μM oligomycin, (B) 40 μM FCCP, and (C) 20/40 μM rotenone/antimycin A. The final results were normalized to total protein levels (BCA protein assay, Visual Protein, ROC) in each well.

### 4.15. Measurement of Progesterone Released from KGN Cells

Oligomycin, an inhibitor of respiration complex ⅤⅤ, was used as a negative control. After the treatment of the KGN cells with different concentrations of aspartame (5, 10, and 20 mM) for 24 and 48 h, the culture media were collected and the progesterone level was detected using a commercial progesterone ELISA kit (Cayman Chemical, Ann Arbor, MI, USA). All procedures were performed following the standard manufacturer’s protocols.

### 4.16. Statistical Analysis

Continuous variables are expressed as the mean ± standard deviation, and categorical variables are expressed as the frequency and percentage. Multivariate regression was performed to examine the relationship between participants’ aspartame intake and infertility outcomes, namely time to conceive, infertility risk, and ART use. Statistical models were adjusted for a priori confounders, namely maternal age, educational level, family income, pre-pregnancy obesity status, and total calorie intake. For the management of missing values, the mean and model were used to fill the missing values of continuous and categorical variables, respectively. All analyses were performed using R version 4.0.3 (R Foundation for Statistical Computing). In animal and cell experiments, Student’s t test was used to compare data between two groups. One-way analysis of variance with Duncan’s post-test was used for multiple comparisons (those among more than three groups). GraphPad Prism 8 software (GraphPad Software, San Diego, CA, USA) was used to obtain figures. A *p* value of <0.05 indicated statistical significance.

## 5. Conclusions

To conclude, although aspartame is largely used as a replacement of traditional sugar to reduce calorie intake, its safety for women’s reproductive health remains unclear. Our data, incorporating clinical, animal, and cell experiments, indicate that aspartame consumption increased the risk of infertility by 1.79-fold by impairing oocyte maturation. Long-term consumption of aspartame altered the estrus cycle and reduced AMH and progesterone secretion, resulting in increased oxidative stress and decreased antioxidative enzymes in the ovary and granulosa cells. These subsequently resulted in compromised mitochondrial function and triggered mitochondrial biogenesis to compensate for mitochondrial dysfunction. Our study indicated the adverse effects of aspartame usage, specifically for women of reproductive age. Women preparing for pregnancy, health practitioners, and dietary guideline recommendations should consider reducing the dosage of aspartame consumption.

## Figures and Tables

**Figure 1 ijms-23-12740-f001:**
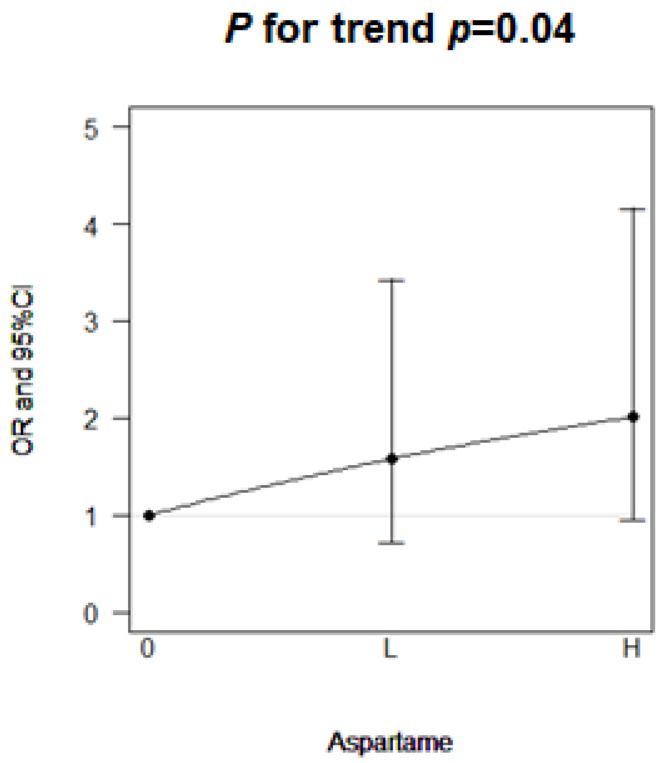
Dose–response effect of aspartame consumption on the risk of infertility in young women (age ≤ 35 years).

**Figure 2 ijms-23-12740-f002:**
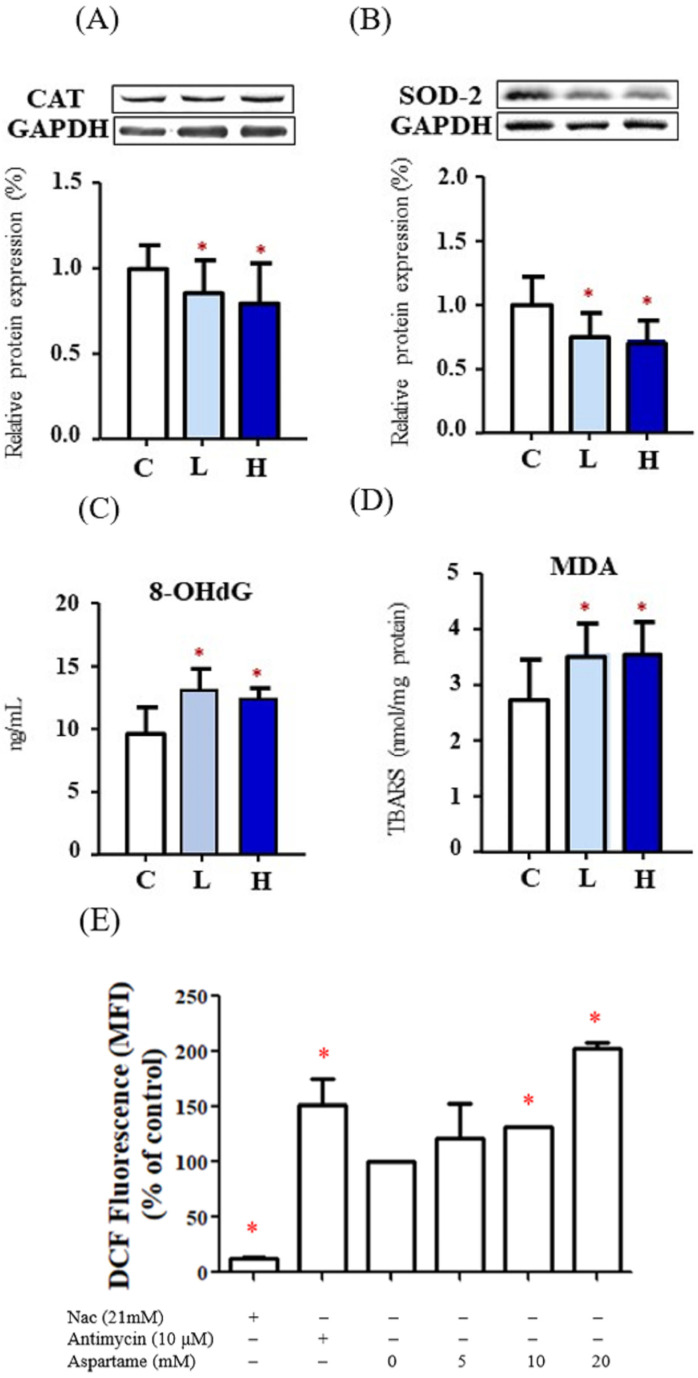
Effect of aspartame treatment on the protein levels of antioxidant enzymes and oxidative stress in rats and cells. Aspartame-treated groups contain lower antioxidant enzymes and higher oxidative stress in ovary and KGN cells. (**A**) The protein levels of catalase (60 kDa), the most abundant and effective enzyme in cytosol (n = 5). (**B**) The protein levels of superoxide dismutase-2 (25 kDa), a mitochondrial antioxidant enzyme (n = 5). The western blot data are normalized with GAPDH so that value of the control is regarded as 1.0. (**C**) 8-OHdG levels, oxidative stress caused DNA damage (n = 5). (**D**) MDA, oxidative stress caused lipid peroxidation (n = 5). (**E**) After KGN cells were treated with 5–20 mM aspartame for 48 h, cells were collected and stained with DCFDA followed by flow cytometry analysis. Quantitative results of ROS are shown in the plot (n = 3). NAC (N-acetyl-l-cysteine) was used as negative control. Antimycin was shown as positive control. All data are expressed as mean ± SD. * *p* < 0.05 as compared with control. C, control. L, low dose of aspartame. H, high dose of aspartame. CAT, catalase. SOD, superoxide dismutase. 8-OHdG, 8-hydroxy-2-deoxyguanosine. MDA, malondialdehyde. TBARS, thiobarbituric acid reactive substances. MFI, medium fluorescence intensity.

**Figure 3 ijms-23-12740-f003:**
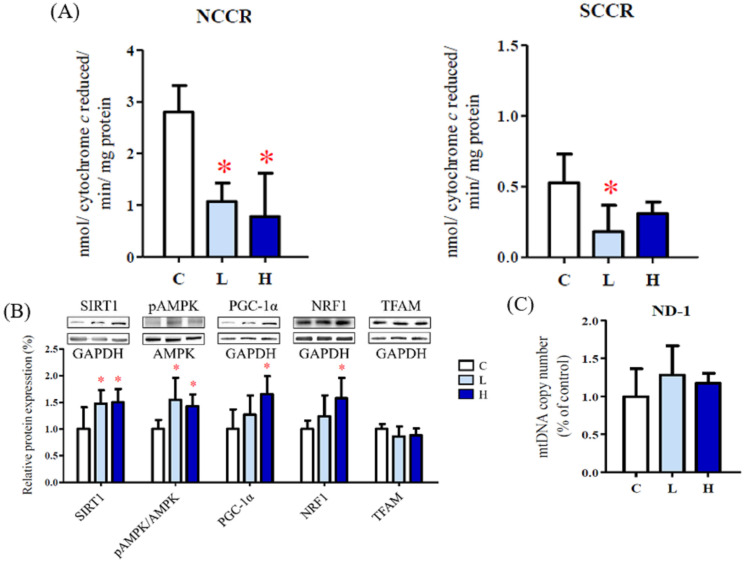
Effect of aspartame treatment on mitochondrial biogenesis and function in rats and cells. Aspartame upregulates mitochondrial biogenesis as a compensatory mechanism to make up for poor mitochondrial function. (**A**) NCCR, mitochondrial complex I and III activities and SCCR, mitochondrial complex II and III activities (n = 5). (**B**) The mitochondrial biogenesis-related protein levels in ovary (n = 5). (**C**) The mitochondrial DNA copy numbers of ovary (n = 5). Data are normalized with GAPDH. Oxygen consumption rate (OCR) was monitored using the seahorse XFe-24 analyzer with the sequential injection of oligomycin, carbonyl cyanide-p-trifluoromethoxy phenylhydrazone (FCCP), and rotenone/antimycin. (**D**) After analyzing, the maximal respiration, spare respiratory capacity and ATP production in KGN cells (n = 3) were calculated. (**E**) The mitochondrial biogenesis-related protein levels in KGN cells treated with 5–20 mM aspartame for 48 h (n = 3). Data are normalized with GAPDH. SIRT1 (120 kDa); AMPK (62 kDa); PGC-1α (95 kDa); NRF1 (54 kDa); TFAM (28 kDa); ATP5α (60 kDa). All data are expressed as mean ± SD. * *p* < 0.05 as compared with control. C, control. L, low dose of aspartame. H, high dose of aspartame. NCCR, nicotinamide adenine dinucleotide–cytochrome c reductase. SCCR, succinate-cytochrome c reductase. SIRT1, NAD-dependent deacetylase sirtuin-1. AMPK, AMP-activated protein kinase. PGC, peroxisome proliferator-activated receptor-gamma coactivator. NRF1, nuclear respiratory factor 1. TFAM, mitochondrial transcription factor A. ND-1, NADH-ubiquinone oxidoreductase chain 1. 0, 0 mM; 5, 5 mM; 10, 10 mM; 20, 20 mM.

**Figure 4 ijms-23-12740-f004:**
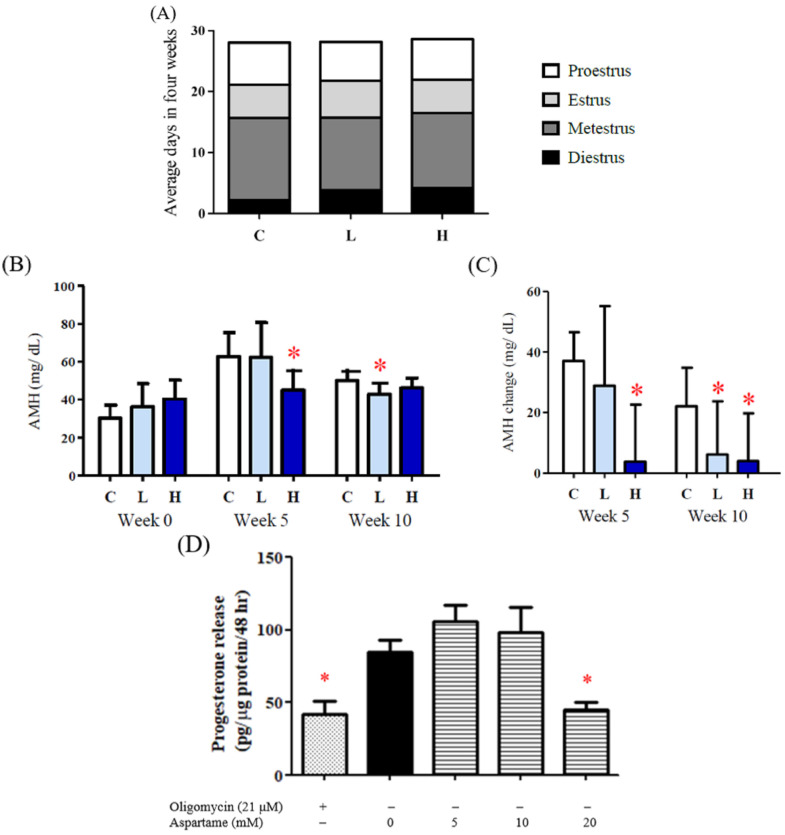
Effect of aspartame treatment on estrus cycle, serum AMH, and progesterone release in rats and cells. Serum AMH and progesterone secretion decreased more in the group treated with aspartame. (**A**) Estrus cycle of different experimental groups after 8–12 weeks intervention. Data were accessed from the result of the vagina smear. (**B**) Serum AMH, a reproductive-related hormone, in weeks 0, 5, and 10. AMH is distinguished from the data from other ages, as it can only be compared with the results in the same age. (**C**) Change of serum AMH in different treatment groups. (**D**) KGN cells treated with 20 mM aspartame significantly decreased progesterone release, which is highly correlated with ATP production. All data are expressed as mean ± SD. * *p* < 0.05 as compared with control. C, control. L, low dose of aspartame. H, high dose of aspartame. AMH, anti-mullerian hormone.

**Table 1 ijms-23-12740-t001:** Clinical characteristics of participants.

Characteristics	Total	
	N	840
	Mean/N	SD/%
Maternal age (years)	33.63	4.08
≤35 years old	472	59.75%
>35 years old	318	40.25%
Gestational age at study (weeks)	23.97	9.86
Gestational age at birth (weeks)	38.63	1.86
Pre-pregnancy BMI (kg/m^2^)	21.37	3.29
Overweight (27 > BMI ≥ 24)	86	10.25%
Obesity (BMI ≥ 27)	51	6.08%
Gestational weight gain (kg)	10.79	5.26
Caloric intake (kcal)	1451.21	516.78
Physical activity		
Low	368	64.34%
Moderate	97	16.96%
High	107	18.71%
Education level		
High school or below	41	4.88%
College or university	598	71.19%
Post-graduate school	201	23.93%
Family income, NTD		
<600,000	231	27.50%
600,001–1,000,000	256	30.48%
>1,000,000	353	42.02%
Smoking status		
Never	837	99.41%
Former	3	0.36%
Time to conceive (months)	12.06	4.08
Infertility *	164	19.52%
Aspartame consumption	208	24.76%

* Infertility was defined as time to conceive longer than 12 months.

**Table 2 ijms-23-12740-t002:** Association of aspartame consumption with the risk of infertility, TMINC (n = 845).

Age Group	Aspartame	Time to Conceive (Months)			Infertility Risk		
		^†^ β	95% CI	*p* Value	OR	95% CI	*p* Value
All	No intake	Ref			Ref		
	Intake	1.28	(−1.34, 3.91)	0.34	1.30	(0.87, 1.93)	0.20
≤35 y/o	No intake	Ref			Ref		
	Intake	1.26	(−1.60, 4.11)	0.39	1.79	(1.00, 3.22)	0.05
>35 y/o	No intake	Ref			Ref		
	Intake	1.39	(−3.16, 5.95)	0.55	1.01	(0.58, 1.75)	0.92

Models were adjusted for maternal age, educational level, family income, pre-pregnancy obesity status, and total calorie intake. ^†^ β indicates the numbers of more months for time to conceive among intake group as compared to no intakes. OR = odds ratio; CI = confidence interval.

**Table 3 ijms-23-12740-t003:** Basic characteristics of the three experimental groups (control, low aspartame, and high aspartame) of Sprague Dawley rats at week 0 and 12.

Characteristics		Control	Low Aspartame	High Aspartame
Food intake (g/kg body weight/day)		69.4 ± 2.2	68.4 ± 2.7	69.2 ± 3.4
Weight gain (g)	Week	83.9 ± 9.2	77.0 ± 14.2	83.1 ± 15.2
Weight (g)	0	213.9 ± 6.30	210.4 ± 7.40	209.2 ± 8.20
	12	297.7 ± 12.1	283.5 ± 13.4	285.9 ± 16.0
Organ weight at week 12				
Heart (g)	OW	1.14 ± 0.15	1.18 ± 0.26	1.15 ± 0.0
	%BW	0.39 ± 0.01	0.40 ± 0.01	0.39 ± 0.00
Liver (g)	OW	7.80 ± 0.26	7.49 ± 0.39	7.53 ± 0.96
	%BW	2.66 ± 0.09	2.56 ± 0.13	2.57 ± 0.33
Kidneys (g)	OW	2.03 ± 0.22	1.96 ± 0.18	1.83 ± 0.23
	%BW	0.69 ± 0.08	0.69 ± 0.08	0.62 ± 0.08
Kidney fat (g)	OW	1.71 ± 0.75	2.56 ± 1.42	3.24 ± 0.88 *
	%BW	0.58 ± 0.26	0.87 ± 0.48	1.10 ± 0.30 *
Inguinal fat (g)	OW	0.78 ± 0.58	1.14 ± 0.46	1.21 ± 0.23
	%BW	0.27 ± 0.20	0.39 ± 0.16	0.41 ± 0.08 *
Ovaries (g)	OW	0.11 ± 0.02	0.11 ± 0.03	0.10 ± 0.02
	%BW	0.037 ± 0.008	0.038 ± 0.011	0.034 ± 0.006
Uterus (g)	OW	0.78 ± 0.41	0.86 ± 0.46	0.73 ± 0.40
	%BW	0.26 ± 0.14	0.29 ± 0.16	0.25 ± 0.14
Uterus fat (g)	OW	1.25 ± 0.50	1.36 ± 0.43	1.42 ± 0.40
	%BW	0.43 ± 0.17	0.47 ± 0.15	0.48 ± 0.14
Serum biochemical	Week			
Glucose (mg/dL)	0	121.46 ± 16.14	118.02 ± 11.03	127.22 ± 14.49
	12	110.38 ± 13.96	125.06 ± 16.48	134.94 ± 13.58 *
Insulin (μU/mL)	0	0.32 ± 0.09	0.26 ± 0.05	0.29 ± 0.04
	12	0.24 ± 0.17	0.28 ± 0.02 *	0.28 ± 0.03
HOMA-IR	0	0.105 ± 0.04	0.08 ± 0.02	0.09 ± 0.02
	12	0.06 ± 0.01	0.08 ± 0.02	0.08 ± 0.01 *
TG (mg/dL)	0	22.21 ± 4.89	23.91 ± 3.01	20.80 ± 4.25
	12	18.22 ± 0.70	17.43 ± 4.37	22.89 ± 5.48 *
AST (U/L)	0	8.76 ± 1.38	7.70 ± 1.45	9.11 ± 1.44
	12	11.03 ± 1.90	11.90 ± 1.81	13.95 ± 2.12 *
ALT (U/L)	0	3.82 ± 0.97	4.84 ± 2.48	3.44 ± 0.83
	12	3.36 ± 0.80	5.07 ± 1.32	7.32 ± 2.60 *

OW = organ weight; BW = body weight. Values are the mean ± SD (n = 5 rats per group), * *p* < 0.05 compared with the control group.

## Data Availability

Not applicable.

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
