# Peer review of "Aspartame Consumption, Mitochondrial Disorder-Induced Impaired Ovarian Function, and Infertility Risk"

_ijms, 2022, doi:10.3390/ijms232112740_

Round 1
Reviewer 1 Report
The work is well written, well-conceived and innovative. the introduction is clear and comprehensive, the purpose is clear the methods are well described and adequate the results are well described and presented and support the initial hypothesis, adequate statistical analysis and well-organized discussions
The question addressed is a very important health issue especially for western population which are facing to severe problems with obesity and infertility. To evaluate the effects of sugar surrogate, which are very often abused, on fertility is really interesting. The topic is relevant and there still some aspects unclear that need to clarified. It add new insight in the molecular pathways modulated by these compounds. The methodology is appropriate but for sure it is always possible to deepening the issue. In my opinion it should be very important to apply RNA-seq for transcriptomic analysis to obtain a complete overview of the molecular pathways affected by these compounds. The conclusions consistent with the evidence and arguments presented and do they address the main question posed. The references are appropriate.
Author Response
Q1: The question addressed is a very important health issue especially for western population which are facing to severe problems with obesity and infertility. To evaluate the effects of sugar surrogate, which are very often abused, on fertility is really interesting. The topic is relevant and there still some aspects unclear that need to clarified. It add new insight in the molecular pathways modulated by these compounds. The methodology is appropriate but for sure it is always possible to deepening the issue. In my opinion it should be very important to apply RNA-seq for transcriptomic analysis to obtain a complete overview of the molecular pathways affected by these compounds. The conclusions consistent with the evidence and arguments presented and do they address the main question posed. The references are appropriate.
A1: Thank you for affirming our manuscript. We agreed with you that to obtain a complete overview of the molecular pathways might requires the RNA seq for transcriptomic analysis. However, in this study, we focused on analyzing the effect of Aspartame-compromised mitochondrial function and triggered mitochondrial biogenesis. Molecular pathways investigated in this study surround the relevant signaling pathways of mitochondria. Moreover, our remaining animal ovaries were not adequate for examining the transcriptomic analysis. We will consider to examine the transcriptomic pathways using RNA seq in future studies. Thank you for your recommendations.
Reviewer 2 Report
Please, see attached file with comments.
